# *Dracocephalum jacutense* Peschkova from Yakutia: Extraction and Mass Spectrometric Characterization of 128 Chemical Compounds

**DOI:** 10.3390/molecules28114402

**Published:** 2023-05-28

**Authors:** Zhanna M. Okhlopkova, Mayya P. Razgonova, Zoya G. Rozhina, Polina S. Egorova, Kirill S. Golokhvast

**Affiliations:** 1Department of Biology, North-Eastern Federal University, Belinsky Str. 58, 677000 Yakutsk, Russia; zhm.okhlopkova@s-vfu.ru (Z.M.O.); zoarozina287@gmail.com (Z.G.R.); 2N.I. Vavilov All-Russian Institute of Plant Genetic Resources, B. Morskaya 42-44, 190000 Saint-Petersburg, Russia; golokhvast@sfsca.ru; 3Institute of Biotechnology, Bioengineering and Food System, Far Eastern Federal University, Sukhanova 8, 690950 Vladivostok, Russia; 4Yakutsk Botanical Garden, Institute for Biological Problems of Cryolithozone Siberian Branch of Russian Academy Sciences, Lenina pr. 41, 677000 Yakutsk, Russia; egorovaps@ibpc.ysn.ru; 5Siberian Federal Scientific Centre of Agro-BioTechnologies of the Russian Academy of Sciences, Centralnaya 2b, 630501 Krasnoobsk, Russia

**Keywords:** *Dracocephalum*, polyphenols, tandem mass spectrometry, ion trap

## Abstract

*Dracocephalum jacutense* Peschkova is a rare and endangered species of the genus *Dracocephalum* of the Lamiaceae family. The species was first described in 1997 and listed in the Red Data Book of Yakutia. Significant differences in the multicomponent composition of extracts from *D. jacutense* collected in the natural environment and successfully introduced in the Botanical Garden of Yakutsk were identified by a team of authors earlier in a large study. In this work, we studied the chemical composition of the leaves, stem, and inflorescences of *D. jacutense* using the tandem mass spectrometry method. Only three cenopopulations of *D. jacutense* were found by us in the territory of the early habitat—in the vicinity of the village of Sangar, Kobyaysky district of Yakutia. The aboveground phytomass of the plant was collected, processed and dried as separate parts of the plant: inflorescences, stem and leaves. Firstly, a total of 128 compounds, 70% of which are polyphenols, were tentatively identified in extracts of *D. jacutense*. These polyphenol compounds were classified as 32 flavones, 12 flavonols, 6 flavan-3-ols, 7 flavanones, 17 phenolic acids, 2 lignans, 1 dihydrochalcone, 4 coumarins, and 8 anthocyanidins. Other chemical groups were presented as carotenoids, omega-3-fatty acids, omega-5-fatty acids, amino acids, purines, alkaloids, and sterols. The inflorescences are the richest in polyphenols (73 polyphenolic compounds were identified), while 33 and 22 polyphenols were found in the leaves and stems, respectively. A high level of identity for polyphenolic compounds in different parts of the plant is noted for flavanones (80%), followed by flavonols (25%), phenolic acids (15%), and flavones (13%). Furthermore, 78 compounds were identified for the first time in representatives of the genus *Dracocephalum*, including 50 polyphenolic compounds and 28 compounds of other chemical groups. The obtained results testify to the unique composition of polyphenolic compounds in different parts of *D*. *jacutense*.

## 1. Introduction

The genus *Dracocephalum* (family Lamiaceae) includes a total of 77 species. They are annual and perennial herbaceous plants, and occasionally dwarf shrubs. The species are native to Europe, Eurasia, North Asia and North America. The genus *Dracocephalum* is of high practical interest due to the accumulation of secondary metabolites, especially polyphenolic compounds, in its vegetative and generative organs. Terpenoids, steroids, flavonoids, alkaloids, lignans, phenols, coumarins, cyanogenic compounds, and glucosides have been identified in the chemical composition of representatives of the genus *Dracocephalum* [1,2,3,4,5]. Some components have antioxidant, antihypoxic, immunomodulatory, and anticancer effects [6,7,8,9].

Many scientific studies have been carried out on the phytochemical composition of representatives of the genus *Dracocephalum* in recent years. Four new (undescribed) terpenoids have been isolated from dried aerial parts of *D. moldavica*, including a monoterpenoid glycoside, an iridoid glycoside, a sesquiterpene and a triterpenoid, as well as nine known terpenoids. The chemical structure of the compounds was established using spectroscopy, HRESIMS data analysis and acid hydrolysis. Of these, five compounds were found in the genus *Dracocephalum* for the first time [10]. The UPLC-Q-TOF-MS method was used to study the qualitative and quantitative composition of secondary metabolites (flavonoids, phenolic acids, and coumarins) in the aerial part of *D. moldavica*, depending on the growth period and geographical location [11]. The study of polyphenolic compounds in *D. moldavica* using LC-MS revealed the content of rosmarinic acid as the main component, in the range of 5.337 ± 0.0411 and 6.320 ± 0.0535 mg/mL [12].

Five species of *Dracocephalum* grow in the territory of Yakutia, which is characterized by a sharply continental climate, close continuous occurrence of permafrost, and snow cover that is preserved for almost seven months a year. Among these species, *D. jacutense* is the only one listed in the Red Data Book of Yakutia [13]. *D. jacutense* grows in stony sparse steppe phytocenoses (Figure 1). To date, only a few cenopopulations of the plant have survived. The comparative analysis of the chemical composition of aerial parts of *D. jacutense* Peschkova collected both in controlled conditions (the Botanical Garden of Yakutia) and in a natural-growth area (the vicinity of the village of Sangar, Kobyaysky district of Yakutia) was performed by a team of authors in a previous large study [14]. A total of 156 bioactive compounds were successfully characterized in extracts of *D. jacutense* based on their accurate MS (Mass Spectrometry) fragment ions by searching online databases and the reported literature. A detailed study of the composition by tandem mass spectrometry revealed a significant difference in the polyphenol composition of the samples.

Wild-grown plant samples had a higher number of polyphenolic compounds (92 compounds) than plant samples grown in the Botanical Garden (56 compounds), which was not previously described in the genus *Dracocephalum*. In addition, a total of 37 compounds of other chemical groups were identified that were not previously identified in the genus *Dracocephalum*. In general, the extract of *D. jacutense* grown in wild conditions was found to be a richer source of flavones, flavanols, flavan-3-ols, phenolic acids, and anthocyanidins than plants grown in controlled conditions in the Botanical Garden.

In general, studies of the phytochemical composition of representatives of the genus *Dracocephalum* are of great importance for determining their potential use in medicine, the development of new drugs and other pharmaceutical industries. The aim of this work is a comparative analysis of the phytochemical profile of various parts of *D. jacutense*, i.e., leaves, inflorescences, and stems, collected in the vicinity of the village of Sangar in the Kobyaysky district of Yakutia during an expedition in July 2022. Maceration extracts of *D. jacutense* were analyzed by ion trap HPLC-MS/MS and showed a greater diversity of chemical compounds present in different parts of the plant. The ion trap was used in the scan range *m*/*z* 100–1700 for MS. A four-stage ion separation mode (MS/MS mode) was implemented. Extracts of plant inflorescences, leaves and stems were analyzed separately. The extracts from *D. jacutense* were analyzed by high-performance liquid chromatography (HPLC) coupled with the ion trap in order to characterize chemical compounds from different parts of *D. jacutense*. The compounds were characterized by interpreting the mass spectrum provided by the ion trap-MS/MS, as well as comparing with information from the literature.

## 2. Results

A total of 128 compounds were tentatively identified in the plant extracts, of which 70% were polyphenols. These polyphenol compounds were classified as 32 flavones, 12 flavonols, 6 flavan-3-ols, 7 flavanones, 17 phenolic acids, 2 lignans, 1 dihydrochalcone, 4 coumarins, and 8 anthocyanidins. Other chemical groups were presented as carotenoids, omega-3-fatty acids, omega-5-fatty acids, amino acids, purines, alkaloids, and sterols.

All the identified compounds, along with MS/MS data, molecular formulas, and their comparative profile for *D. jacutense*, are summarized in Table A1 (Appendix A). Of the identified compounds, 70% are polyphenols, and 30% are amino acids, fatty acids, purine, alkaloid, sterol, carotenoids, etc. Compounds of the polyphenol group were represented in inflorescences by 73 variations, in leaf extracts by 33 compounds, and in stem extracts by 22 polyphenols.

Of these compounds, 78 were identified for the first time in the genus *Dracocephalum;* 50 were polyphenolic compounds and 28 were from other chemical groups (amino acids, fatty acids, triterpenic acids, etc.). Furthermore, 36 polyphenolic compounds and compounds of other chemical groups (fatty acids, naphthoquinone, pterocarpan, amino acids, triterpenic acids, zeaxanthin, etc.) were found for the first time in extracts from the inflorescences, while 6 polyphenolic compounds were found for the first time in leaf extracts, and 2 polyphenolic compounds were found in stem extracts. Figure A1, Figure A2 and Figure A3 (from Appendix A) below show ion chromatograms separately for extracts from inflorescences, stems, and leaves of *D. jacutense*.

The greatest similarity in the identified chemical compounds is found in representatives of the genera *Mentha*, *Vaccinium*, *Rosmarinus*, *Astragali*, and *Eucalyptus*. In addition, Rhodioloside C (monoterpene glycoside), previously described in *Rhodiola rosea*, was found in leaf extracts [15,16,17] and *Rhodiola crenulata* [18].

The newly identified polyphenols belonged to nine classes, including 11 phenolic acids and their conjugates, 14 flavones, 6 flavonols, 4 flavan-3-ols, 3 flavanone, 5 anthocyanins, 2 lignans, 4 coumarins, and 1 dihydrochalcone (Table 1). Newly identified compounds from other chemical groups belonged to 11 classes, including 1 benzenediol, 3 amino acid and their conjugates, 2 fatty amides, 3 omega-3 fatty acids, 1 omega-5 fatty acid, 4 carotenoids, 1 monoterpene glycoside, 1 diterpenoid naphthoquinone, 4 triterpenic acids, 1 pterocarpan, 1 dihydrochalcone, and others.

### 2.1. Flavones

#### 2.1.1. 7-Hydroxy(iso)flavones

The flavones formononetin (compound **1**), and calycosin [3′-Hydroxyformononetin] (compound **4**) have already been characterized as a component of *Astragali Radix* [19,20,21], Huolisu Oral Liquid [22], and the Chinese herbal formula Jian-Pi-Yi-Shen pill [23]. The flavone formononetin and calycosin were found in extracts from leaves of *D. jacutense.* The CID-spectrum in positive ion mode of flavone calycosin from extracts of leaves of *D. jacutense* is shown in Figure 2.

The [M + H]^+^ ion produced two fragment ions at *m*/*z* 253.27 [aglycone-CH_3_OH] and *m*/*z* 167.21 (Figure 2). The fragment ion with *m*/*z* 253.3 yielded two daughter ions at *m*/*z* 209.33 and *m*/*z* 135.36. It was identified in the bibliography in extracts of *Astragali radix* [19,20,21] and Huolisu Oral Liquid [22]. The CID-spectrum in positive ion mode of formononetin from extracts of leaves of *D. jacutense* is shown in Figure 3.

The [M + H]^+^ ion produced six fragment ions at *m*/*z* 213.3, *m*/*z* 199.35, *m*/*z* 185.29, *m*/*z* 161.24, *m*/*z* 133.33, and *m*/*z* 117.3 (Figure 3). The fragment ion for *m*/*z* 213.3 yielded four daughter ions at *m*/*z* 169.21, *m*/*z* 157.26, *m*/*z* 143.24, and *m*/*z* 129.29. The fragment ion for *m*/*z* 169.21 yielded two daughter ions at *m*/*z* 143.27 and *m*/*z* 129.33. It was identified in the bibliography in extracts of *Astragali radix* [19,20,21], Huolisu Oral Liquid [22] and the Chinese herbal formula Jian-Pi-Yi-Shen pill [23]. The base peak ion chromatogram in positive ion mode and base peak ion chromatogram in negative ion mode of *D. jacutense* (experiment 2484) are shown in Figure 4. 

#### 2.1.2. Dihydroxyflavones

The flavones genkwanin (compound **5**) and Dihydroxy-dimethoxy(iso)flavone (compound **10**) have already been characterized as a component of *D. palmatum* [1], *Astragali radix* [20], *Rosmarinus officinalis* [24], propolis [25], etc. These flavones were found in extracts from leaves and flowers of *D. jacutense.* The CID-spectrum in positive ion mode of genkwanin from extracts of leaves of *D. jacutense* is shown in Figure 5.

The [M + H]^+^ ion produced three fragment ions at *m*/*z* 270, *m*/*z* 242, and *m*/*z* 167 (Figure 5). The fragment ion for *m*/*z* 270 yielded daughter ions at *m*/*z* 242. The fragment ion for *m*/*z* 242 yielded daughter ions at *m*/*z* 213, *m*/*z* 197, and *m*/*z* 124. It was identified in the bibliography in extracts of *D. palmatum* [1,5], *Rosmarinus officinalis* [24], and *Menthae Haplocalycis* [26]. The base peak ion chromatogram in positive ion mode and base peak ion chromatogram in negative ion mode of *D. jacutense* (experiment 2490) are shown in Figure 6. 

#### 2.1.3. Trihydroxyflavones

The flavones apigenin (compound **2**), diosmetin (compound **7**), and chrysoeriol (compound **8**) have already been characterized as a component of *D. palmatum* [1], *Dracocephalum* [5,14], propolis [25], *D. moldavica* [27], *Rhus coriaria* [28], etc. The flavones diosmetin, and chrysoeriol were found in extracts from the leaves of *D. jacutense,* and the flavone apigenin was found in extracts of the flowers of *D. jacutense.*

#### 2.1.4. Hexahydroxyflavone

The flavone myricetin (compound **12**) has already been characterized as a component of *Vaccinium macrocarpon* [29] and Andean blueberry [30]. This flavone was found in extracts from inflorescences of *D. jacutense.*


### 2.2. Flavan-3-ols

The flavan-3-ols catechin (compound **46**), (epi)catechin (compound **47**), gallocatechin (compound **48**), catechin-3-*O*-gallate (compound **49**), and epigallocatechin-3-gallate (compound **50**) have already been characterized as a component of *Dracocephalum* [1,5,14], *Sanguisorba officinalis* [31], *C. edulis* [32], *and Camellia kucha* [33]. The flavan-3-ol catechin-3-*O*-gallate was found in extracts from leaves of *D. jacutense.*


## 3. Discussion

The polyphenol composition distribution table is shown in Table A2 (Appendix B). The comparison table shows the presence of some flavonoids in all three types of extracts, including the polyphenols acacetin, luteolin, cirsimaritin, luteolin 7-*O*-glucoside, kaempferol, astragalin, kaempferol-3-*O*-glucuronide, naringenin, eriodictyol, prunin, eriodictyol 7-*O*-glucoside, rosmarinic acid, and caffeic acid derivative. The results of the research turned out to be more representative, finding 73 polyphenols in extracts from inflorescences, 33 polyphenols in extracts from leaves and 22 polyphenols in extracts from stems of *D. jacutense.*

The analysis shows that the overwhelming presence of the polyphenolic group was found in the inflorescence of *D. jacutense*. Moreover, the majority of this group of flavonoids are flavones, amounting to 21 chemical compounds, 29% of the total compounds of the polyphenol group. In second place in terms of the number of identified polyphenol groups are hydroxybenzoic and hydroxycinnamic acids, amounting to 15 chemical compounds, 21% of the total compounds. In third place in terms of the number of detected compounds are flavonols, amounting to 12 chemical compounds, 16% of the total amount of polyphenols.

It should be noted that some of the chemical compounds found in *D. jacutense* were first tentatively identified in the genus *Dracocephalum*. These include the polyphenol compounds formononetin, calycosin, cirsimaritin, 5,7-dimethoxyluteolin, myricetin, cirsiliol, taxifolin-3-*O*-hexoside, catechin 3-*O*-gallate, epigallocatechin-3-gallate, ferreirin, homoeriodictyol, salvianic acid, protocatechuic acid-*O*-hexoside, etc.

Figure 7 shows a Venn diagram built on the data obtained during the mass spectrometric study of the presence of polyphenols in different parts of the plant. The Venn diagram data shows that 13 compounds (14.6%) are present in all three parts of the plant, 8 polyphenolic compounds (9%) are present in both the inflorescences and in the leaves, and 4 polyphenolic compounds (4.5%) are present in both the inflorescences and in the stems of the plant.

A detailed interpretation of the identified compounds in inflorescences, leaves, and stems of *D. jacutense* is presented in Table 2. 

## 4. Materials and Methods

### 4.1. Plant Material

Separate parts (leaves, stems, inflorescences) of *D. jacutense* Peschkova were collected during expedition work in the territory of the Kobyaysky district of Yakutia from July 14 to 19 July 2022 (Figure 8). The aboveground phytomass was collected at the stage of full flowering of the plant. A few seeds were at the stage of milky ripeness and were husked (extracted) from inflorescences during office processing before drying the phytomass. All samples were morphologically authenticated according to the current standard of the State Pharmacopoeia of the Russian Federation [34]. 

### 4.2. Chemicals and Reagents

HPLC-grade acetonitrile was purchased from Fisher Scientific (Southborough, UK), and MS-grade formic acid was obtained from Sigma-Aldrich (Steinheim, Germany). Ultrapure water was prepared using a Siemens Ultra Clear system (Siemens Water Technologies, Gunzburg, Germany), and all other chemicals were analytical grade.

### 4.3. Fractional Maceration 

Fractional maceration (repeated infusion) provides for a change in the concentration difference at the phase boundary due to the renewal of the extractant. In this case, the amount of the extractant is divided into portions, and the infusion time is divided into periods. [35]. From 300 g of the sample, 10 g of inflorescences, leaves, and stems were randomly selected for maceration. The total amount of the extractant (ethyl alcohol of reagent grade) was divided into three parts, and the parts of plant were consistently infused in the first, second, and third parts. The solid–solvent ratio was 1:20. The infusion of each part of the *D. jacutense* samples continued for 7 days at room temperature.

### 4.4. Liquid Chromatography

A Shimadzu LC-20 Prominence HPLC Pump (Shimadzu, Kyoto, Japan) equipped with a UV sensor and C18 silica reverse phase column (4.6 × 150 mm, particle size: 2.7 μm) was used to perform the separation of multicomponent mixtures. The gradient elution program with two mobile phases (A, deionized water; B, CH_3_CN with formic acid 0.1% *v*/*v*) was as follows: 0, 0–4 min, 100% CH_3_CN; 4–60 min, 100–25% CH_3_CN; 60–75 min, 25–0% CH_3_CN; control washing 50–60 min, 100% A. The entire HPLC analysis was performed with a UV–VIS detector SPD-20A (Shimadzu, Kyoto, Japan) at a wavelength of 230 nm for identification compounds, a temperature of 50 °C, and a total flow rate of 0.25 mL min^−1^. The liquid chromatography equipment was combined into one line with an ion trap amaZon SL (Bruker Daltoniks, Bremen, Germany) for the identification of biologically active compounds.

### 4.5. Mass Spectrometry

The chemical compounds were identified by comparing their mass spectra, mass spectrometry fragmentation, and retention time with a home-library database built by the Food Products Group at the Far East Federal University (Russian Federation), based on data from other spectroscopic equipment and data from scientific literature. MS analysis was performed on an ion trap amaZon SL (Bruker Daltonics, Germany) equipped with an ESI source in negative and positive ion modes. The optimized parameters were as follows: ionization source temperature, 70 °C; gas flow, 4 L/min; nebulizer gas (atomizer), 7.3 psi; capillary voltage, 4500 V; end plate bend voltage, 1500 V; fragmentary, 280 V; collision energy, 60 eV. 

## 5. Conclusions

In total, 128 chemical compounds were identified in the extracts of the rare species *D. jacutense*, which grows only in the environs of the village of Sangar, the Kobyaysky district of Yakutia, using HPLC-MS/MS with an ion trap and database comparison. Of these, 73 polyphenolic compounds were found in extracts from inflorescences, 33 in extracts from leaves, and 22 in extracts from stems. Of the total number of polyphenols found, 14% of the compounds are found in all types of extracts. These include four flavones, three flavanols, four flavanones and two phenolic acids. A large share of the identity for polyphenolic compounds in different parts of *D. jacutense* is noted for flavanones, for which the identity is 80%, then for flavonols (25%), phenolic acids (15%), and flavones (13%).

Thus, in terms of the individuality of the classes of polyphenolic compounds in *D. jacutense*, it can be noted that flavonoids, isoflavanone, phenylpropanoic acid, hydroxycinnamic acids, lignans, hydroxycoumarins, coumarins, and coumarin glucoside are found only in inflorescences, while hydroxybenzoic acid and dihydrochalcone are found only in stems.

All obtained data testify to the unique phytochemical composition of extracts from different parts of *D. jacutense*. This plant species is characterized by a narrow local distribution; at present, only three cenopopulations have been preserved in the sparse steppe phytocenoses of the Kobyaysky district of Yakutia. 

## Figures and Tables

**Figure 1 molecules-28-04402-f001:**
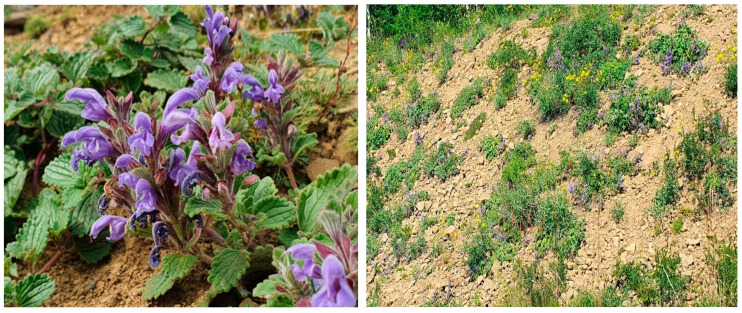
*D. jacutense* Peschkova (Kobyaysky district of Yakutia, photo taken by Rhozina, July 2022).

**Figure 2 molecules-28-04402-f002:**
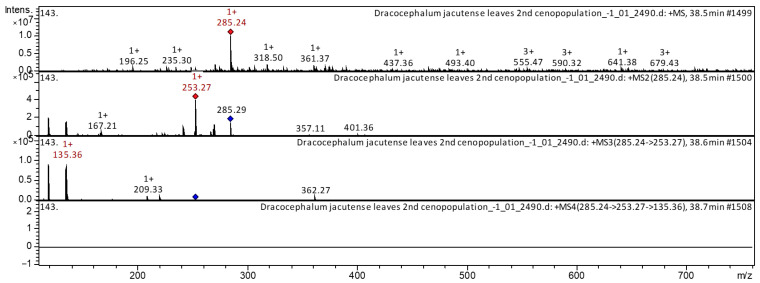
CID-spectrum of calycosin [3′-Hydroxyformononetin] from extracts of leaves of *D. jacutense*, at *m*/*z* 285.24.

**Figure 3 molecules-28-04402-f003:**
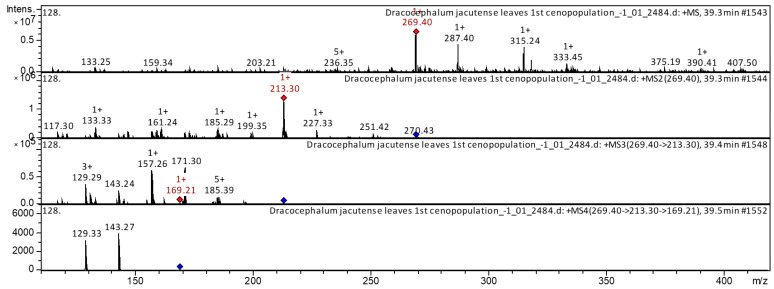
CID-spectrum of formononetin from extracts of leaves of *D. jacutense*, at *m*/*z* 269.4.

**Figure 4 molecules-28-04402-f004:**
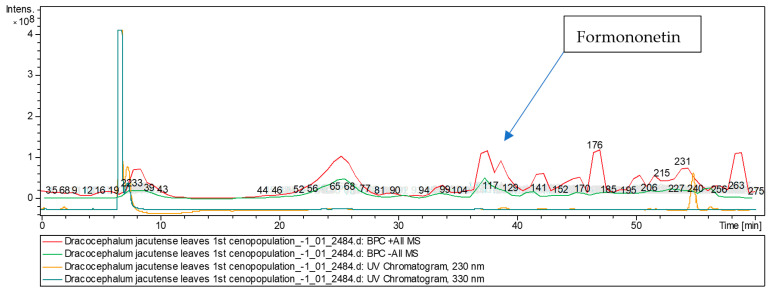
Base peak ion chromatogram in positive ion mode and base peak ion chromatogram in negative ion mode of *D. jacutense* (experiment 2484).

**Figure 5 molecules-28-04402-f005:**
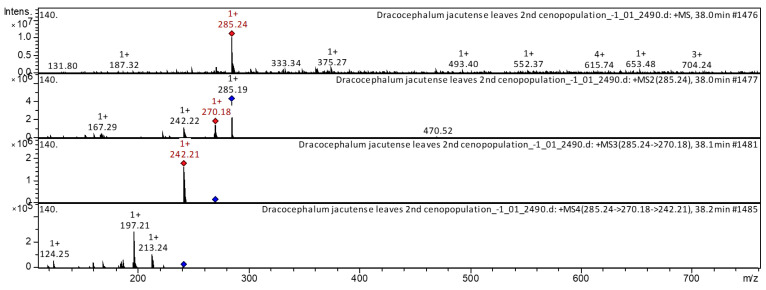
CID-spectrum (experiment 2490) of genkwanin from extracts of leaves of *D. jacutense*, at *m*/*z* 285.

**Figure 6 molecules-28-04402-f006:**
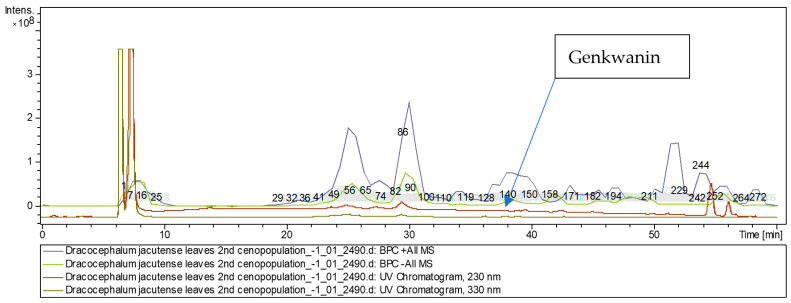
Base peak ion chromatogram in positive ion mode and base peak ion chromatogram in negative ion mode of *D. jacutense* (experiment 2490).

**Figure 7 molecules-28-04402-f007:**
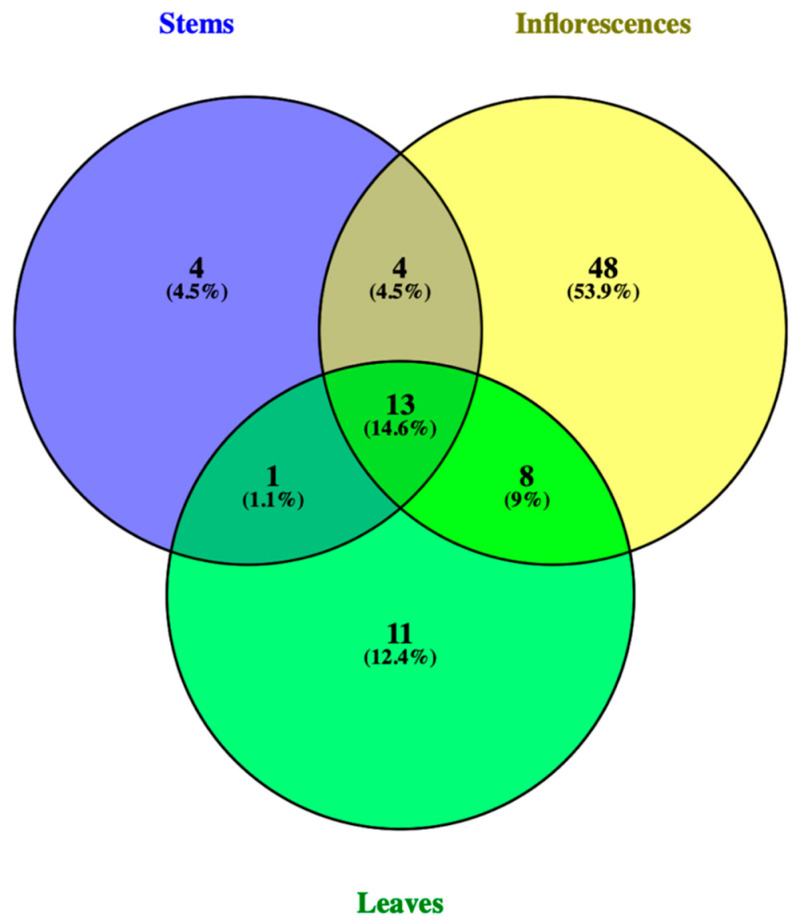
Venn diagram representing a study of the polyphenolic composition of compounds in the inflorescences, leaves, and stems of *D. jacutense*.

**Figure 8 molecules-28-04402-f008:**
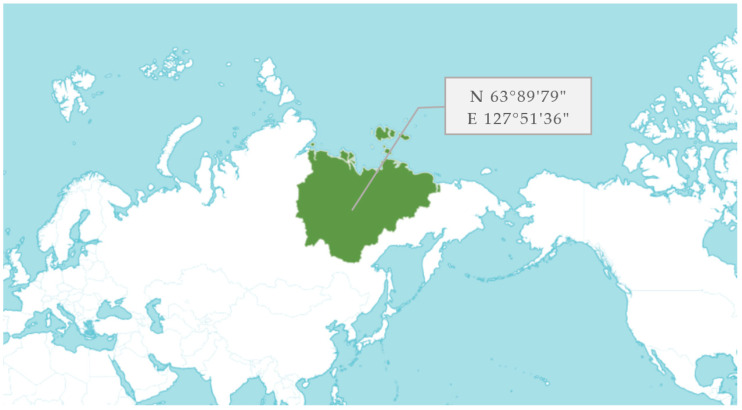
Collection areas of *D. jacutense* Peschkova in the territory of the Kobyaysky district of Yakutia (Russian Federation).

**Table 1 molecules-28-04402-t001:** Polyphenols identified in the extracts of *D. jacutense* in positive and negative ionization modes using HPLC-ion trap-MS/MS.

No	Class of Compound	Identified Polyphenol	Formula
**1**	Flavone	**Formononetin** [Biochanin B; Formononetol] *	**C_16_H_12_O_4_**
**2**	Flavone	**Apigenin** [5,7-Dixydroxy-2-(40Hydroxyphenyl)-4H-Chromen-4-One]	**C_15_H_10_O_5_**
**3**	Flavone	**Acacetin** [Linarigenin; Buddleoflavonol]	**C_16_H_12_O_5_**
**4**	Flavone	**Calycosin** [3′-Hydroxyformononetin] *	**C_16_H_12_O_5_**
**5**	Flavone	**Genkwanin** [Gengkwanin; Puddumetin; Apigenin 7-Methyl Ether]	**C_16_H_12_O_5_**
**6**	Flavone	**Luteolin**	**C_15_H_10_O_6_**
**7**	Flavone	**Diosmetin** [Luteolin 4′-Methyl Ether; Salinigricoflavonol]	**C_16_H_12_O_6_**
**8**	Flavone	**Chrysoeriol** [Chryseriol]	**C_16_H_12_O_6_**
**9**	Flavone	**Cirsimaritin** *	**C_17_H_14_O_6_**
**10**	Flavone	**Dihydroxy-dimethoxy(iso)flavone** *	**C_17_H_14_O_6_**
**11**	Flavone	**5,7-Dimethoxyluteolin** *	**C_17_H_14_O_6_**
**12**	Flavone	**Myricetin** *	**C_15_H_10_O_8_**
**13**	Flavone	**Isothymusin**	**C_17_H_14_O_7_**
**14**	Flavone	**Cirsiliol** *	**C_17_H_14_O_7_**
**15**	Flavone	**Dimethoxy-trihydroxy(iso)flavone** *	**C_17_H_14_O_7_**
**16**	Flavone	**Nevadensin**	**C_18_H_16_O_7_**
**17**	Flavone	**Gardenin B** [Demethyltangeretin] *	**C_19_H_18_O_7_**
**18**	Flavone	**5-Hydroxy-6,7,8,3′,4′-pentamethoxyflavone** *	**C_20_H_20_O_8_**
**19**	Flavone	**Apigenin *O*-hexoside**	**C_21_H_20_O_10_**
**20**	Flavone	**Apigenin-7-*O*-glucoside** [Apigetrin; Cosmosiin]	**C_21_H_20_O_10_**
**21**	Flavone	**Apigenin 7-*O*-glucuronide**	**C_21_H_18_O_11_**
**22**	Flavone	**Acacetin 7-*O*-glucoside** [Tilianin]	**C_22_H_22_O_10_**
**23**	Flavone	**Luteolin 7-*O*-glucoside** [Cynaroside; Luteoloside]	**C_21_H_20_O_11_**
**24**	Flavone	**Acacetin 7-*O*-** ***β*-d-glucuronide**	**C_22_H_20_O_11_**
**25**	Flavone	**6,4′-Dimethoxyisoflavone-7-*O*-glucoside** *	**C_23_H_24_O_10_**
**26**	Flavone	**Diosmetin-7-*O*-** ***β*-glucoside**	**C_22_H_22_O_11_**
**27**	Flavone	**Apigenin-*O*-rhamnoside** *	**C_22_H_22_O_11_**
**28**	Flavone	**Chrysoeriol-7-*O*-glucuronide** *	**C_22_H_20_O_12_**
**29**	Flavone	**Acacetin 7-** ***β*-*O*-(6″-acetyl)-glucoside**	**C_24_H_24_O_11_**
**30**	Isoflavone	**Apigenin 7-*O*-** ** *β* ** ** - ** ** d ** **-(6″-*O*-** **malonyl)-glucoside**	**C_24_H_22_O_13_**
**31**	Flavone	**Acacetin 7-*O*-** ** *β* ** **-d-(6″-*O*-malonylated)-glucoside**	**C_25_H_24_O_13_**
**32**	Flavone	**Chrysoeriol *O*-hexoside *C*-hexoside** *	**C_28_H_32_O_16_**
**33**	Flavonol	**Kaempferol**	**C_15_H_10_O_6_**
**34**	Flavonol	**Quercetin**	**C_15_H_10_O_7_**
**35**	Flavonol	**Dihydroquercetin** (Taxifolin; Taxifoliol)	**C_15_H_12_O_7_**
**36**	Flavonol	**Isorhamnetin** *	**C_16_H_12_O_7_**
**37**	Flavonoid	**3,5-Diacetyltambulin** *	**C_22_H_20_O_9_**
**38**	Flavonol	**Astragalin** [Kaempferol 3-*O*-glucoside; Astragaline]	**C_21_H_20_O_11_**
**39**	Flavonol	**Quercitrin** [Quercetin 3-*O*-rhamnoside; Quercetrin] *	**C_21_H_20_O_11_**
**40**	Flavonol	**Kaempferol-3-*O*-glucuronide**	**C_21_H_18_O_12_**
**41**	Flavonol	**Taxifolin-3-*O*-hexoside** [Dihydroquercetin-3-*O*-hexoside] *	**C_21_H_22_O_12_**
**42**	Flavonol	**Kaempferol 3-*O*-rutinoside**	**C_27_H_30_O_15_**
**43**	Flavonol	**Kaempferol-3,7-Di-*O*-glucoside** *	**C_27_H_30_O_16_**
**44**	Flavonol	**Kaempferol dihexoside rhamnoside** *	**C_33_H_40_O_20_**
**45**	Flavan-3-ol	**(epi)Afzelechin** *	**C_15_H_14_O_5_**
**46**	Flavan-3-ol	**Catechin** [D-Catechol] *	**C_15_H_14_O_6_**
**47**	Flavan-3-ol	**(epi)catechin**	**C_15_H_14_O_6_**
**48**	Flavan-3-ol	**Gallocatechin** [+(−)Gallocatechin]	**C_15_H_14_O_7_**
**49**	Flavan-3-ol	**Catechin 3-*O*-gallate** *	**C_22_H_18_O_10_**
**50**	Flavan-3-ol	**Epigallocatechin-3-gallate** *	**C_22_H_18_O_11_**
**51**	Flavanone	**Naringenin** [Naringetol; Naringenine]	**C_15_H_12_O_5_**
**52**	Flavanone	**Eriodictyol** [3′,4′,5,7-tetrahydroxy-flavanone]	**C_15_H_12_O_6_**
**53**	Isoflavanone	**Ferreirin** *	**C_16_H_14_O_6_**
**54**	Trihydroxyflavanone	**Homoeriodictyol** *	**C_16_H_14_O_6_**
**55**	Flavanone	**Prunin** [Naringenin-7-*O*-glucoside]	**C_21_H_22_O_10_**
**56**	Flavanone	**Eriodictyol-7-*O*-glucoside** [Pyracanthoside; Miscanthoside]	**C_21_H_22_O_11_**
**57**	Flavanone	**Eriodictyol-7-*O*-glucuronide** *	**C_21_H_20_O_12_**
**58**	Hydroxycinnamic acid	***p*-Coumaric acid** *	**C_9_H_8_O_3_**
**59**	Hydroxycinnamic acid	**3,4-Dihydroxyhydrocinnamic acid** *	**C_9_H_10_O_4_**
**60**	Phenolic acid	**2,3,4,5-Tetrahydroxybenzoic acid** *	**C_7_H_6_O_6_**
**61**	Phenolic acid	**Salvianic acid A** [Danshensu] *	**C_9_H_10_O_5_**
**62**	Hydroxybenzoic acid	**Ellagic acid** [Benzoaric acid; Elagostasine; Lagistase; Eleagic acid]	**C_14_H_6_O_8_**
**63**	Phenolic acid	**Protocatechuic acid-*O*-hexoside** *	**C_13_H_16_O_9_**
**64**	Phenolic acid	**Caffeic acid-4-*O*-*β*-d-hexoside** [Caffeoyl-*O*-hexoside]	**C_15_H_18_O_9_**
**65**	Phenolic acid	**Chlorogenic acid** [3-*O*-Caffeoylquinic acid]	**C_16_H_18_O_9_**
**66**	Phenolic acid	**Isochlorogenic acid** *	**C_16_H_18_O_9_**
**67**	Phenolic acid	**Rosmarinic acid**	**C_18_H_16_O_8_**
**68**	Phenolic acid	**Caffeic acid derivative**	**C_16_H_18_O_9_Na**
**69**	Phenolic acid	**1/3/4/5-*p*-Coumaroylquinic acid * + C_2_H_2_O**	**C_18_H_20_O_9_**
**70**	Phenolic acid	**8,8′-Aryl-Diferulic acid** *	**C_20_H_18_O_8_**
**71**	Phenolic acid	**Caffeic acid hexoside dimer** *	**C_31_H_40_O_17_**
**72**	Phenolic acid	**Salvianolic acid B** [Danfensuan B] *	**C_36_H_30_O_16_**
**73**	Phenylpropanoic acid	**Sagerinic acid**	**C_36_H_32_O_16_**
**74**	Phenolic acid	**Clerodendranoic acid H** *	**C_36_H_32_O_16_**
**75**	Lignan	**Phillygenin** [Sylvatesmin; Phyllygenol; Forsythigenol] *	**C_21_H_24_O_6_**
**76**	Lignan	**Medioresinol** *	**C_21_H_24_O_7_**
**77**	Dihydrochalcone	**Phloretin** [Dihydronaringenin; Phloretol] *	**C_15_H_14_O_5_**
**78**	Hydroxycoumarin	**Umbelliferone** [Skimmetin; Hydragin] *	**C_9_H_6_O_3_**
**79**	Coumarin	**Fraxetin** [7,8-Dihydroxy-6-methoxycoumarin] *	**C_10_H_8_O_5_**
**80**	Hydroxycoumarin	**Umbelliferone hexoside** *	**C_15_H_16_O_8_**
**81**	Coumarin glycoside	**Fraxin** [Fraxetin-8-*O*-glucoside] *	**C_16_H_18_O_10_**
**82**	Anthocyanidin	**Petunidin**	**C_16_H_13_O_7+_**
**83**	Anthocyanidin	**Pelargonidin-3-*O*-glucoside (callistephin)**	**C_21_H_21_O_10_**
**84**	Anthocyanidin	**Cyanidin-3-*O*-glucoside** [Cyanidin 3-*O*-beta-d-Glucoside; Kuromarin]	**C_21_H_21_O_11+_**
**85**	Anthocyanidin	**Cyanidin 3,5-*O*-diglucoside** *	**C_27_H_31_O_16_**
**86**	Anthocyanidin	**Peonidin-3,5-diglucoside** [Peonin; Peonidin 3-Glucoside-5-Glucoside] *	**C_28_H_33_O_16_**
**87**	Anthocyanidin	**Cyanidin-3-*O*-rutinoside-5-*O*-glucoside** *	**C_33_H_41_O_20_**
**88**	Anthocyanidin	**Delphinidin 3-*O*-rutinoside-5-*O*-glucoside** *	**C_33_H_41_O_21_**
**89**	Anthocyanidin	**Malonyl-shisonin** *	**C_39_H_39_O_21+_**

* Polyphenols identified for the first time in genus *Dracocephalum.*

**Table 2 molecules-28-04402-t002:** Detailed interpretation of the identified compounds in inflorescences, leaves, and stems of *D. jacutense*.

Names	Total	Elements
Inflorescences Leaves Stems	13	Prunin; Kaempferol-3-*O*-glucuronide; Naringenin; Eriodictyol; Rosmarinic acid; Caffeic acid derivative; Luteolin 7-*O*-glucoside; Luteolin; Acacetin; Eriodictyol-7-*O*-glucoside; Cirsimaritin; Kaempferol; Astragalin;
Inflorescences Stems	4	Apigenin-7-*O*-glucoside; Apigenin; Acacetin 7-*O*-glucoside; Homoeriodictyol;
Leaves Stems	1	Diosmetin;
Inflorescences Leaves	8	Petunidin; Fraxetin; Isorhamnetin; Genkwanin; Gallocatechin; Apigenin 7-*O*-beta-d-(6″-*O*-malonyl)-glucoside; Catechin; Cyanidin-3-*O*-glucoside;
Stems	4	Phloretin; Acacetin 7-beta-*O*-(6″-acetyl)-glucoside; 1/3/4/5-p-Coumaroylquinic acid; Ellagic acid;
Inflorescences	48	3,4-Dihydroxyhydrocinnamic acid; Epigallocatechin-3-gallate; Chrysoeriol-7-*O*-glucuronide; Delphinidin 3-*O*-rutinoside-5-*O*-glucoside; Protocatechuic acid-*O*-hexoside; Pelargonidin-3-O-glucoside; Eriodictyol-7-*O*-glucuronide; Cyanidin-3-*O*-rutinoside-5-*O*-glucoside; Quercetin; Diosmetin-7-*O*-beta-glucoside; Ferreirin; Quercetrin; (epi)Afzelechin; Kaempferol-3,7-Di-*O*-glucoside; Fraxin; Apigenin 7-*O*-glucuronide; 3,5-Diacetyltambulin; 2,3,4,5-Tetrahydroxybenzoic acid; Salvianic acid A; Apigenin *O*-hexoside; Caffeic acid hexoside dimer; Cirsiliol; Salvianolic acid B; Chlorogenic acid; (epi)catechin; Apigenin-*O*-rhamnoside; Acacetin 7-*O*-beta-d-glucuronide; Cyanidin 3,5-*O*-diglucoside; Umbelliferone; Medioresinol; Malonyl-shisonin; 8,8′-Aryl-Diferulic acid; Phillygenin; p-Coumaric acid; Kaempferol dihexoside rhamnoside; 6,4′-Dimethoxyisoflavone-7-*O*-glucoside; Sagerinic acid; Taxifolin-3-*O*-hexoside; Caffeic acid-4-*O*-beta-d-hexoside; Umbelliferone hexoside; Clerodendranoic acid H; Myricetin; Chrysoeriol *O*-hexoside C-hexoside; 5,7-Dimethoxyluteolin; Isochlorogenic acid; 5-Hydroxy-6,7,8,3′,4′-pentamethoxyflavone; Dihydroquercetin; Kaempferol 3-*O*-rutinoside;
Leaves	11	Gardenin B; Nevadensin; Peonidin-3,5-diglucoside; Isothymusin; Chrysoeriol; Formononetin; Calycosin; Dihydroxy-dimethoxy(iso)flavone; Acacetin 7-*O*-beta-d-(6″-*O*-malonylated)-glucoside; Catechin 3-*O*-gallate; Dimethoxy-trihydroxy(iso)flavone;

The polyphenol composition distribution of *D. jacutense* is summarized in Table A2 (Appendix B). It should be noted that some of the chemical compounds found in *D. jacutense* were first tentatively identified in the genus *Dracocephalum*. These include the polyphenol compounds formononetin, calycosin, cirsimaritin, 5,7-dimethoxyluteolin, myricetin, cirsiliol, taxifolin-3-*O*-hexoside, catechin 3-*O*-gallate, epigallocatechin-3-gallate, ferreirin, homoeriodictyol, salvianic acid, protocatechuic acid-*O*-hexoside, etc.

## Data Availability

Not applicable.

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
