# Peer review of "Dracocephalum jacutense Peschkova from Yakutia: Extraction and Mass Spectrometric Characterization of 128 Chemical Compounds"

_molecules, 2023, doi:10.3390/molecules28114402_

Round 1

Reviewer 1 Report

Please, refer to pdf file.

Author Response

Dear Reviewer.

Thank you very much for your work and your time.

We have tried to carefully all the errors and inaccuracies that you pointed out.

Sincerely yours,

Dr. Razgonova

Reviewer 2 Report

I believe that it would be easier to follow the results, if instead of the chromatograms (they can be relocated in the appendices) there was a table with the identified compounds. At this moment the results part looks like a count of the identified compounds, combined with an evaluation of the spectral behavior for some of them.

In the case of discussions, a formulation of a vegetable product (parts of the plant) with different therapeutic uses would be useful. The analytical part is complex, therefore a more detailed interpretation of the obtained results would be useful.

Rows 215-219, respectively 234-238 are identical. In table 1, the use of the measurement unit after each retention time is redundant. Authors should include it in the head of the table.

In line 50, 55, Dracocephalum should be written in italics. In row 50 is preferable, for a better understanding of the text, replacing the digit 4 with text (four). In the case of the mass detector, it is correctly written with capital letters Q-TOF (row 55).

The article needs extensive English editing, which makes it difficult to read.

Author Response

Dear Reviewer.

We express our gratitude to you for the work on the text and the time spend.

We tried to correct all found errors and shortcomings and supplemented the text.

Sincerely yours

Round 2

Reviewer 2 Report

Dear authors,

I don't think that the information in your study should be duplicated. Table 1 in the Appendix fully reproduces Table 1 from the main text. Under these conditions, I consider that the information in Table 1 should possibly be centralized in a simplified manner, or exemplified by the classes of compounds described in the text, with the primary substances.

I think the header row from the tables should be repeated on each page to make it easier to follow the spectral parameters.

Otherwise, your material is valuable and suitable for publication.

Author Response

Dear Reviewer.

I have corrected all the errors and inaccuracies that you pointed out. Indeed, the tables repeated each other, I modified the first table and it is of a brief informative nature, and the large table remained, as it was in the Appendix.

Thank you very much for your work and your valuable time.